# Tracked Wall-Climbing Robot for Calibration of Large Vertical Metal Tanks

**Xianlei Chen [1], Yiping Wu [2], Huadong Hao [1], Haolei Shi [1] and Haocai Huang [2,*]**

[1] Zhoushan Institute of Calibration and Testing for Quality and Technology Supervision, Zhoushan 316021, China

[2] Ocean College, Zhejiang University, Zhoushan 316021, China

* Correspondence: hchuang@zju.edu.cn; Tel.: +86-580-2092203



**Featured Application: The tracked wall-climbing robot is used for the capacity calibration of several oil tanks located in Zhoushan city, China. And we also plan to use the robot for the inspection like non-destructive testing of the oil tanks by adding the corresponding sensors.**

**Abstract:** Large vertical metal tanks are the primary vessels for the storage and turnover of crude oil, and the accuracy of their capacity calibrations are of great significance. The optical reference line method (ORLM) is used for capacity calibration and is time-consuming, labor-intensive, and hazardous, because of the elevated work. This paper aims to present a robot to overcome the problems above. We propose a tracked wall-climbing robot (TWCR) with permanent magnetic adhesion tracks, a collapsible scale, and an optional shovel-like rust remover that enable the TWCR to move stably on tank surfaces and perform the ORLM. Two sets of field tests (internal ORLM and external ORLM) indicate that capacity calibration by the TWCR is time saving, convenient, and safe, in addition to being accurate and reliable.

**Keywords:** wall-climbing; robot; calibration; optical reference line method

## 1. Introduction

With the increasing demand for energy and the rapid worldwide growth of the petroleum industry, large vertical metal tanks and various other types of storage tanks, which serve as the basic vessels for storage and turnover of crude oil, have been constructed in large quantities [1]. The tanks undergo mandatory capacity calibrations at set intervals or after maintenance according to the requirement of Chinese national standard JJG 168-2005 (Verification Regulation of Vertical Metal Tank Capacity Sector/Industry) and American Petroleum Institute (API) standard. The accuracy of the capacity calibrations is related not only to the trade settlement, cost accounting, and energy consumption of the domestic companies, but also to the trade fairness between countries and the credibility of national measurement. For example, the throughput of oil, natural gas, and their products at ports above designated size in China was 1.066 billion tons in 2018 [2] and, if there was a 0.1% error in capacity calibration, the measurement error would be a million tons.

The radial deviation measurement is one of the primary steps of the capacity calibration of the large vertical metal oil tanks, which can be performed through several methods, as shown in Table 1.

**Table 1.** Advantages and disadvantages of different methods.

| Method | Advantage | Disadvantage | References |
|---|---|---|---|
| Liquid Fill Method | Direct and very accurate | Time-consuming excessively | JJG 168-2005 |
| Strapping Method | Accurate | Scaffolds is needed, lots of aerial work | API 2.2A |
| Optical Reference Line Method (ORLM) | Accurate, easy to operate relatively, low requirements for equipment | A small amount of aerial work | API 2.2B |
| Electro Optical Distance Ranging Method (EODR) | Automated, fewer operators are needed | Not suitable for rusty surface, increased ranging error with an increase of the measuring elevation angle. | API 2.2D |

The liquid fill method and the strapping method, which are very time-consuming and inconvenient, are used only on small tanks or as a reference. The electro optical distance ranging method (EODR) is wildly used because of its convenience, but when the tanks are built closer, the EODR instrument must be placed near the tanks, which leads to the large measuring elevation angle, as well as the large ranging error. The optical reference line method (ORLM) is considered as one of the main methods in Chinese national standard JJG 168-2005 and API standard chapter 2.2B, because of its accuracy and wide applicability.

In the ORLM, the tank is divided into horizontal and vertical stations, as is shown in Figure 1. The reference circumference is determined by the strapping method. Each ring has two vertical stations which are 1/4 or 3/4 below the horizontal weld seam to obtain its averaged deviation from the reference. In each horizontal station, the magnetic trolley moves vertically and stops at every vertical station where the reference deviation is recorded by the optical device. Finally, the reference circumference and the deviations are used for the calculation of the volume.

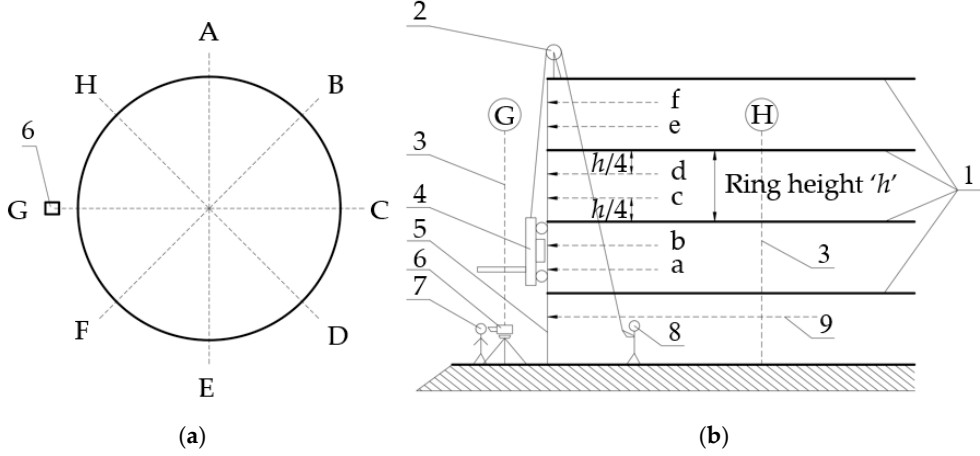

(a)  (b)

**Figure 1.** Working principle of the optical reference line method (ORLM): (**a**) Top view of the tank (**b**) side view of the tank (A–H horizontal stations, a–f vertical stations, 1—horizontal weld seams, 2—pulley, 3—optical reference line, 4—magnetic trolley, 5—tank surface, 6—optical device, 7—operator #1, 8—operator #2, 9—reference circumference).

There remain a number of drawbacks in the conventional ORLM:

1. This method requires 2–3 persons to spend more than 2 h to perform the task, and one of them has to install the pulleys at the top of the tanks (more than 20 m in the large tanks) which poses a significant risk to the operator.
2. At least one operator is required to pull the rope to lift the trolley and maintain its balance manually. As there could be as many as 200 measuring stations, it could lead to operator fatigue.

3. When the trolley is moving vertically to reach the measuring stations, its location can only be confirmed visually, which results in deviations at large elevation angles.

The conventional ORLM has exhibited great potential for the applications of wall-climbing robots (WCR), which could solve the above problems.

A number of types of WCRs capable of working in the vertical plane with different adhesion and locomotion mechanisms have been studied. Several major adhesion mechanisms are identified as six types. The magnetic adhesion employs permanent magnetization or electrical magnetization for climbing [3–7]. The suction/propulsion adhesion generates a pressure difference for climbing on the smooth surface [8,9]. The mechanical grasping adhesion utilizes grasping mechanisms like a bio-inspired spine or tiny hook for climbing [10,11]. The chemical adhesion makes use of adhesive tapes on the robot feet to climb on the vertical surface [12]. The thermoplastic adhesion employs a kind of special material which provides temperature-dependent adhesion forces for climbing [13], and the electrostatic force is also studied for surface adhesion [14]. With respect to the locomotion mechanisms, six categories are classified, which are arms and legs [15,16], wheels [4–6], tracks [3,7], wires and rails [17], sliding frame [18] and sandwich configuration [19]. Tables 2 and 3 show the comparisons of the adhesion and locomotion mechanisms, respectively.

**Table 2.** Comparison of adhesion mechanisms.

| Type | Payload Capacity | Limitation | References |
|---|---|---|---|
| Magnetic | High | Ferromagnetic surfaces only | [3–7] |
| Suction and Propulsion | High | Not suitable for rough surfaces | [8,9] |
| Mechanical Grasping | High | Not suitable for smooth and flat surfaces | [10,11] |
| Chemical Adhesion | Low | Vulnerable to dust | [12] |
| Thermoplastic Adhesion | Low | Slow state-changing process, trace left | [13] |
| Electrostatic Adhesion | Low | Low robustness | [14] |

**Table 3.** Comparison of locomotion mechanisms.

| Type | Payload Capacity | Speed | Limitation | References |
|---|---|---|---|---|
| Arms and Legs | Medium | Slow | Complex mechanism | [15,16] |
| Wheels | Medium | Fast | Small contact face | [4–6] |
| Tracks | High | Fast | Low steering capacity | [3,7] |
| Wires and Rails | High | Fast | Wires or rails needed | [17] |
| Sliding Frame | High | Medium | Discontinuous locomotion | [18] |
| Sandwich | High | Fast | Two carts needed to climb the nonferromagnetic surface | [19] |

Typically, the design of the adhesion and locomotion mechanisms, which determine the payload, moving velocity, and energy supply, vary according to the working conditions and working content. In a number of applications where the WCRs are designed for heavy work, such as rust-removal [3,20], wall cleaning [8,9,17], and automatic welding [4,21], the robots are optimized for high loads and high energy consumption to ensure the working progress. The locomotion and adhesion mechanisms of these robots could be large-sized and heavy, and a number of types of locomotion and adhesion mechanisms could even be combined to meet specific requirements. On the other hand, different design concepts when designing WCRs for inspection and measuring, such as portability, low energy consumption, and inspection capacity, could be of more practical benefit to researchers.

The WCRs used for capacity calibration or inspection of large vertical metal tanks are types of WCR designed specifically for inspecting and measuring. As the walls of the tanks are made of steel, permanent magnets, with the advantage of significant adhesion capacity without power, are widely

used. Magnetic wheel-type mechanisms are used to obtain flexible mobility [22]. However, their contact face is small, which means that a part of the magnetic unit is far from the metal surface, and the adhesion force is correspondingly small. Track-type mechanisms are preferred for their greater contact surface and stable adhesion force. However, it is difficult for them to change direction, and this could cause damage to the walls. A non-contact magnetic adhesion mechanism was developed to obtain simultaneous flexible mobility and greater adhesion forces, but its mechanism is complex and requires additional magnets, which increases the weight [5].

In this study, a robot with climbing ability is proposed to replace the manually-lifted trolley used in the traditional optical vertical line method. As the robot is expected to move in only one direction during the calibration of a single horizontal station, the steering capability is not a high priority. In addition, considering their lightness and reliability, track-type locomotion and permanent magnet adhesion mechanisms are suitable for the calibration of large vertical metal tanks. The scale is designed to be collapsible to avoid colliding with the fire pipelines. Considering that the surface of the tanks could be rusty, a shovel-like rust remover can be mounted to remove the rust and improve the contact between the adhesion mechanism and the surface.

This study is organized as follows: Section 2 presents an overview of the tracked wall-climbing robot (TWCR) and introduces its primary characteristics, including the mechanical architecture, control system and the capacity calibration system, intended for the capacity calibration of large vertical metal tanks; the prototype, field tests and result are discussed in Section 3; and Section 4 presents the conclusions.

## 2. Design of Tracked Wall-climbing Robot (TWCR)

### 2.1. System Overview

The TWCR is designed to replace the magnetic trolley used in the ORLM, which has to be drawn manually to move vertically. It shall have six basic functions:

1.  A scale shall be equipped for the ORLM.
2.  A wired remote control shall be used to control the vertical mobile speed of the robot and minor displacements of the scale. The former gets the robot close to the target location, and the latter allows precise adjustment of the position of the scale.
3.  The linear motion deviation shall be less than 1°50′ from the perpendicular to guarantee that the scale is visible by the optical device during the calibration of a single horizontal station.
4.  The magnetic force shall be sufficiently strong to ensure the safe adhesion and the stability of operation. The gravity of the cables shall be taken into consideration. The robot also needs an extra payload capacity of 1 kg, which allows the operator to install additional equipment, such as probes and monitors, if required.
5.  The robot shall be well adapted to poor surface conditions of tanks.
6.  This robot shall be at an advantage in terms of its small size and lightweight, which means it is portable and able to maneuver through any narrow spaces.

The mechanical structure of the TWCR is shown in Figure 2. The track-type locomotion mechanism and the permanent magnet adhesion mechanism enable the TWCR to move stably on the vertical surface of the tanks. The driving, transmission, and steering mechanism provides power and, with the help of the attitude sensor and the microcontroller, achieves the required linear deviation. The scale is used for the radial deviation measurement in the ORLM.

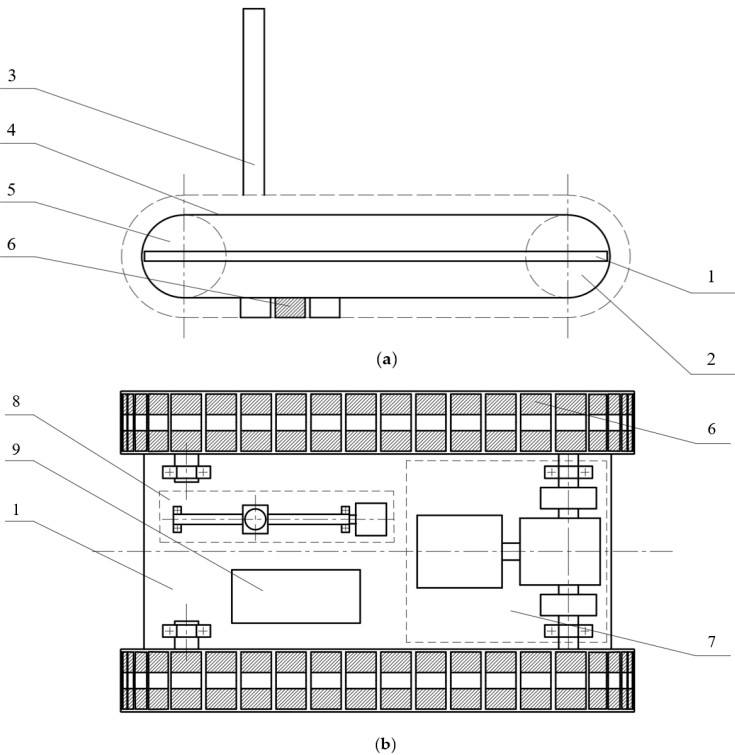

(a)

(b)

**Figure 2.** Mechanical structure of the tracked wall-climbing robot (TWCR): (**a**) Right view (**b**) top view (1—baseboard, 2—driving wheel, 3—scale, 4—tracks, 5—driven wheel, 6—permanent magnetic units, 7—driving, transmission, and steering mechanism, 8—scale moving, and collapsing mechanism, 9—microcontroller).

*2.2. Design of Permanent Magnetic Adhesion and Tracked Locomotion Mechanism*

The TWCR uses permanent magnets to adhere to the metal surface of the tanks and moves forward and backward by the tracked locomotion mechanism, which provides adhesion stability, and the adhesion mechanism remains working in the event of unexpected power failures.

The adhesion mechanism, shown in Figure 3, is designed based on the theoretical analysis and a number of previous studies [23]. The material of the magnet is sintered NdFeB 52SH. A single magnet unit comprises two pieces of permanent magnet and a yoke. The tracks use two aluminum plates to fix the permanent magnet units and protect them from potential damage by the rust, which could be dislodged and block the track.

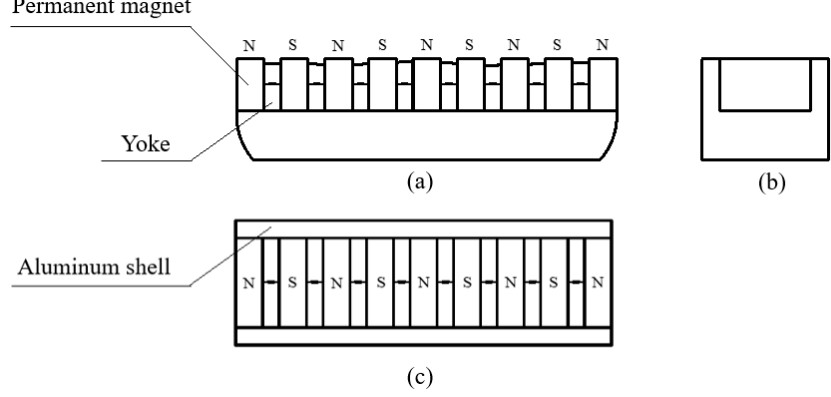

(a)

(b)

(c)

**Figure 3.** Structure of adhesion mechanism: (**a**) Left view; (**b**) front view; (**c**) bottom view.

Previous studies [7,24] have pointed out that the main risks encountered by the TWCR during operation are sliding along the wall and rolling around the lowest contact point the track and the wall. A force analysis model is built to determine the necessary adhering force. In Figure 4, O is the center of mass, P is the lowest contact point, $h$ is the distance between the center of mass and the wall, $l$ is the distance of the highest and lowest contact point, $F_f$ is the friction force between the track and the wall, $G$ is the total weight of the TWCR and the cables, $F_m$ is the adhesion force of each permanent magnetic unit, $n$ is the number of the permanent magnetic unit in contact with the wall and $N_i$ is the supporting force of the wall to each permanent magnetic unit. Every magnetic unit on the track bears the same adhering force, but the supporting force differs, due to the torque of the gravity.

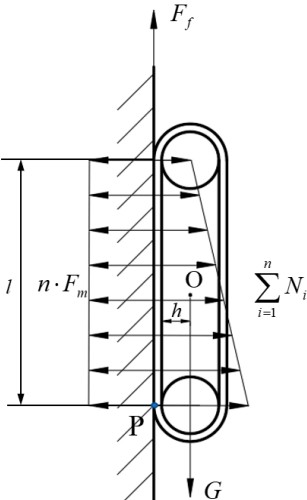

**Figure 4.** Static force analysis of the TWCR.

The necessary friction force that the robot needs to avoid sliding along the wall is given by:

$$F_f \geq G \tag{1}$$

We have,

$$F_f = \sum_{i=1}^{n} \mu N_i = \mu \sum_{i=1}^{n} N_i, \tag{2}$$

where $\mu$ is the friction coefficient. We also have,

$$nF_m = \sum_{i=1}^{n} N_i. \tag{3}$$

According to Equations (1−3), we have,

$$F_m \geq \frac{G}{n\mu}. \tag{4}$$

The adhesion force also has to be great enough to avoid overturning around point P. As the torque at point P cannot be transmitted by the flexible chain of the track, the first upper magnetic unit will be lifted if the overturning happens at point P, the torque balance equation is

$$\frac{1}{2}Gh = (F_m - N_1)l \tag{5}$$

where $1/2Gh$ is the 1/2 torque (as there are two tracks share the load) of the gravity, $(F_{m1} - N_1) \cdot l$ is the torque of the first upper magnetic unit, and we have,

$$N_1 > 0, \tag{6}$$

if the first upper magnetic unit is still able to adhere to the wall, then, according to Equations (5) and (6), we have,

$$F_m \geq \frac{Gh}{2l} \tag{7}$$

According to Equations (4) and (7), we have the necessary condition for the stable adhesion of the TWCR:

$$F_m \geq \max\left(\frac{G}{n\mu}, \frac{Gh}{2l}\right). \tag{8}$$

The weight of the TWCR is about 10 kg according to the evaluation of the 3-D modeling tool (SolidWorks Version 2015, Dassault Systèmes SolidWorks Corp., Waltham, MA, USA), and the weight of the suspension cable (about 30 m) is 3 kg, $G = (10 + 3) \times 9.8 = 127.4$ N. The number of the permanent magnetic unit in contact with the wall (both side) $n$ is 26, the friction coefficient $\mu$ is 0.1–0.2, the height of the center of the mass $h$ is 31.2 mm according to the evaluation of the 3-D modeling tool, the distance of the highest and lowest contact point $l$ is 212 mm. Hence, we have,

$$F_m \geq \max\left(\frac{127.4}{26 \times 0.1}, \frac{127.4 \times 31.2}{2 \times 212}\right) = 49 \text{ N}. \tag{9}$$

Because of the anti-corrosion coating on the tank surface, the gap formed between the permanent magnetic units and the tank surface will inevitably affect the adhesion capacity of the adhesion mechanism. The relation between the length of the gap and the adhesion force of a single magnetic unit was tested, and the result is shown in Table 4. Typically, the thicknesses of tank wall coatings or oxide layers, which should be taken into consideration, are approximately 1 mm. Therefore, the adhesion force generated by a single magnetic unit is approximately 80.6 N which reach the requirement of (9). Therefore, the adhesion capacity is great enough to prevent the robot from sliding along the wall or rolling around the lowest contact point the track and the wall.

**Table 4.** The relation between the length of the gap and the adhesion force of a single magnetic unit.

| Length of Air Gap/mm | 0 | 0.5 | 1.0 | 2.0 | 3.0 | 4.0 | 5.0 |
|---|---|---|---|---|---|---|---|
| Adhesion force /N | 244.1 | 150.2 | 80.6 | 43.5 | 29.6 | 20.2 | 18 |

### 2.3. Design of Driving, Transmission, and Steering Mechanism

Capacity calibrations are conducted before the first use or after maintenance, when the tanks are empty and free from the risk of explosion. Thus, the robot is powered by a wire delivering 220 V AC, which is readily available close to the tanks. An AC servo motor is used as the main power unit of the robot, because of its small size and lightweight. The total torque necessary for both sides that the motor outputs through the reducer are given by:

$$T \geq Gh + 2(M_1 - M_2), \tag{10}$$

where $T$ is the necessary torque that the motor outputs through the reducer, $M_1$ and $M_2$ is the torque of the first upper magnetic unit which is engaging with the surface and the lowest magnetic unit which is disengaging with the surface, respectively. $M_1$ and $M_2$ are changing during the motion of the TWCR, and as shown in Figure 5, we have,

$$M_1 - M_2 < M_1 = F_m l_1 < F_m l_0, \tag{11}$$

where $F_m$ is the adhesion force of each permanent magnetic unit, $l_1$ is the horizontal distance of the center of the permanent magnetic unit and the center of the driven wheel and $l_0$ is the length of the permanent magnetic unit, $l_0 = 15$ mm. Hence, we have,

$$T \geq 127.4 \times 0.0312 + 2 \times (80.6 \times 0.015) = 6.4 \text{ Nm} \tag{12}$$

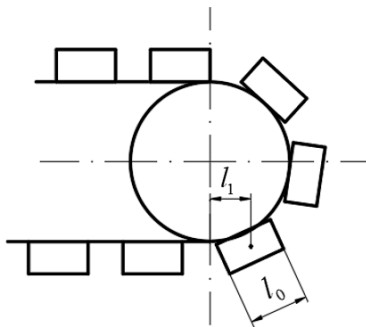

**Figure 5.** The position of the lowest magnetic unit which is disengaging with the surface.

As can be seen in Figure 6, a 30 W AC gear motor with a reduction ratio of 1:36 is installed on the baseboard and delivers power to the gear train comprising two electromagnetic clutches. The rated torque of the motor is 0.35 Nm, and the transmission efficiency of the clutches are 95%, thus, we have,

$$T_1 = 0.35 \times 36 \times 0.95 = 10.26 \text{ Nm} > 6.4 \text{ Nm} \tag{13}$$

which means that the motor meets the requirement of the Equation (12).

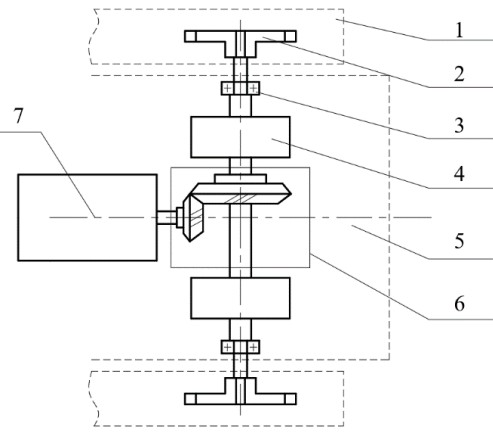

**Figure 6.** Driving, transmission, and steering mechanism (1—track, 2—driving wheel, 3—pillow block, 4—electromagnetic particle clutch, 5—baseboard, 6—bevel gear reducer, 7—motor).

Compared to the use of two motors for traction, a single motor with two clutches, though the presence of clutches introduces extra power consumption and transmission loss, make sufficient room for the calibration system on the compact body. The torque transmitted by the electromagnetic particle clutch is linear with the control current, which means the torque can be easily controlled. When the torque transmitted by the two clutches is the same, the TWCR moves straight, and when there is a torque difference, the wall-climbing robot will steer.

### 2.4. Design of Capacity Calibration System

The TWCR, based on the ORLM, is developed primarily for the capacity calibration of vertical metal tanks. In the ORLM, a trolley that carries a scale is supposed to move up and down vertically

and stop at a number of pre-assigned positions, allowing the operator to record the reference offsets by reading the scale through the optical device. Although the TWCR is designed to replace the manually-lifted trolley, a scale is still required. A secondary 25 W AC gear motor with a reduction ratio of 1:72, which adjusts the position of the scale through a ball screw, is also installed on the baseboard. In practice, the tracked locomotion mechanism moves the robot close to the target location, and adjustments by the scale-moving mechanism obtain the precise position.

To avoid collision between the fire control pipes/ribs arranged above the tank surface and the towering scale, a collapsible scale, tensioned by an elastic band to maintain its straightness, was developed. As can be seen in Figure 7, the upper and lower sections of the scale are connected by a shaft, and an elastic band is used to assist the scale in recovering from a collapsed state. No new structure is introduced, and the scale-moving mechanism is used to fulfill the collapsing function. When the scale approaches one side of the limit, the upper part of the scale is stopped by a bar installed on the outer frame of the robot and the lower part of the scale continues moving forward. This results in the scale folding, as shown in Figure 8, allowing the robot to pass any restricted places.

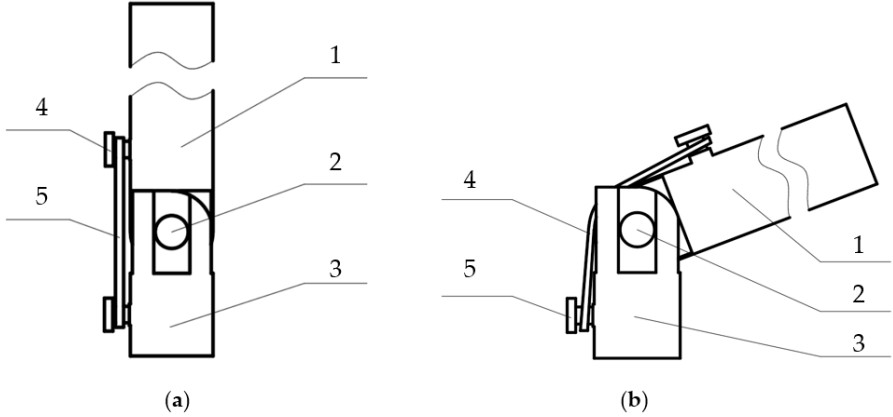

**Figure 7.** The mechanical structure of scale (**a**) the stand state; (**b**) the folded state (1—upper section, 2—shaft, 3—lower section, 4—elastic band, 5—fixing bolt).

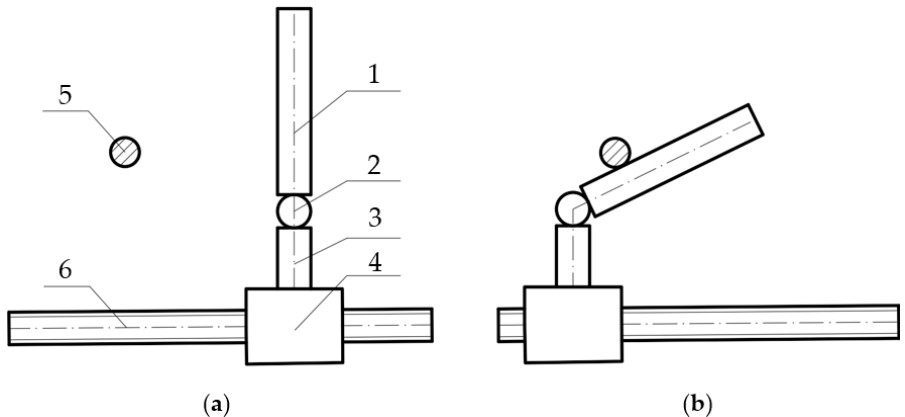

**Figure 8.** Collapsing process of scale (**a**) the stand state (**b**) the folded state (1—upper section, 2—shaft, 3—lower section, 4—base, 5—stopping bar, 6—ball screw).

Considering that the surface of the metal tanks could be significantly rusted, and that the rust could scrape against the permanent magnetic tracks and stain them, which will increase the measured offsets, as well as decrease the adhesion force, a shovel-like rust remover can be installed on the front of the robot. However, one side effect of this rust remover is that it could increase resistance to movement. A rotating mechanism driven by a small motor can lift the rust remover to avoid the problem, as shown in Figure 9.

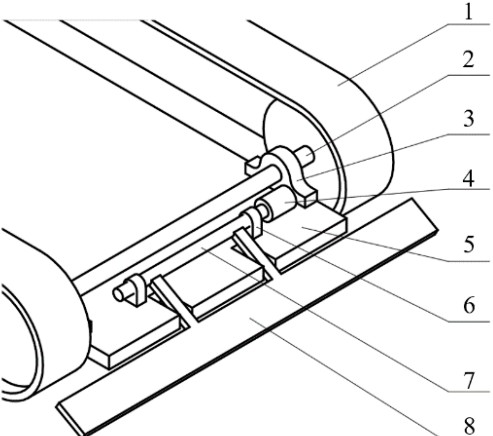

**Figure 9.** Structure of rust remover (1—track, 2—main shaft, 3—main pillow block, 4—small motor, 5—baseboard, 6—small pillow block, 7—small shaft, 8—plank).

## 2.5. Control System

As is shown in Figure 10, the TWCR is controlled by the operator through the wired remote control, and the microcontroller, which receives the control signals and attitude signals from the sensor, outputs the control signals to the main driving motor, electromagnetic particle clutch, driving motors of the rust remover and the scale. The height of the TWCR is visually controlled by the operator according to the distance from the adjacent horizontal weld seams, and the precise position of the scale is measured by the optical device and manually adjusted by the operator through the scale-moving mechanism.

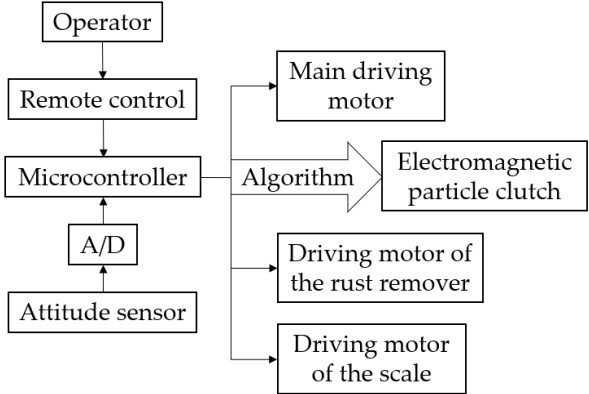

**Figure 10.** Configuration of the control system.

The control algorithm, which is based on the proportional–integral–derivative (PID) algorithm, helps the operator to adjust the trajectory of the TWCR to fulfil the requirement of the linear motion deviation. As is shown in Figure 11, the input of the PID controller is the offset value of the expected attitude and the actual attitude. And the output of the PID controller is the differences of the actuating current for the two electromagnetic clutches, which leads to the torque differences of the two tracks of the robot. Thus, the TWCR steers and the attitude bias is eliminated. The overshoot of the PID controller is suppressed to achieve driving stability, despite the corresponding long adjustment time.

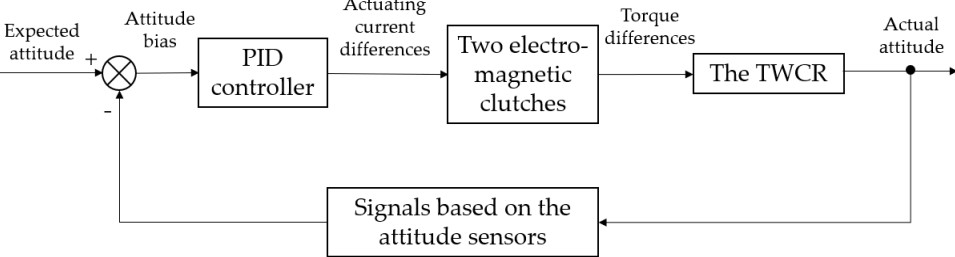

**Figure 11.** Diagram of the proportional–integral–derivative (PID) controller.

## 3. Prototype and Field Tests

### 3.1. Prototype

Using the design described above, the prototype TWCR, shown in Figure 12, was produced, and the details of the tracks and adhesion mechanisms are shown in Figure 13. It has the dimensions of 280 mm × 190 mm × 65 mm (without the scale) and the weight of 9.5 Kg.

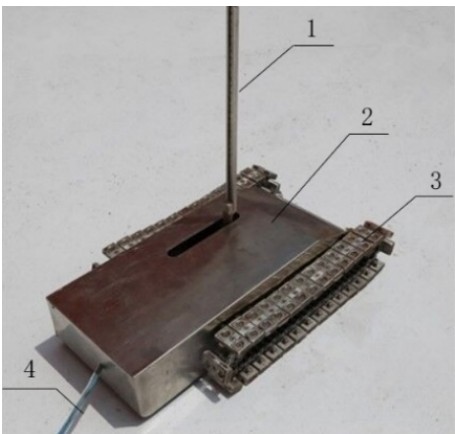

**Figure 12.** Overview of TWCR (1—scale, 2—main part of the robot, 3—tracks, 4—power supply cable).

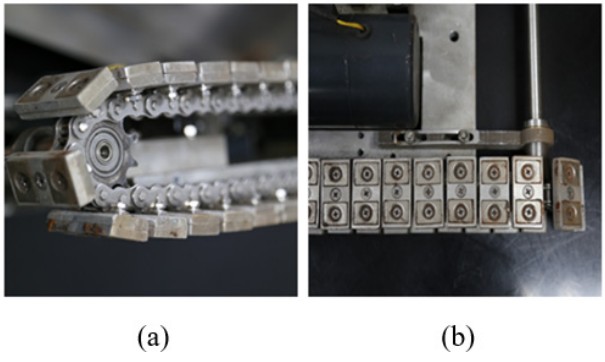

(a)　　　　　　　　　　　　　(b)

**Figure 13.** The details of tracks and adhesion mechanisms; (**a**) side view; (**b**) top view.

As mentioned above, the TWCR is developed to replace the trolley used in the ORLM-based capacity calibration of vertical metal tanks. Two sets of experiments (internal ORLM and external ORLM) were conducted to verify the reliability and accuracy of the capacity calibration using the robot, and to compare the convenience of the prosed method with the traditional method of manually lifting the trolley. At the beginning of the experiment, the robots were carefully placed on the metal surface and aligned vertically, after which the TWCR itself further adjusted the attitude bias (if any).

### 3.2. Internal Measurement Test

The working principles of the internal ORLM were introduced in Section 1. As is shown in Figure 14, an external floating roof tank with a nominal volume of 50,000 m$^3$, located in Zhoushan City, China, was selected for the internal experiments. On-site environmental conditions were wind speed 18 km/h, temperature 28 °C, and relative humidity 65%, in line with the requirement of the ORLM. As a reference, the optical reference line measurement was dependent upon the reference circumference (188,459 mm in this tank) determined by manual strapping of the third course, 1/4 course height below the top of the course horizontal weld seam. The volume of the bottom, the height of the courses, and the volume of the appurtenances were identical for both volume tables. This tank is divided into 60 horizontal stations and 20 vertical stations. The offsets measured by the two methods are presented in Table 5. The radial deviation in the table is the average of the deviations of the 60 horizontal stations in the same ring (each ring has two vertical stations).

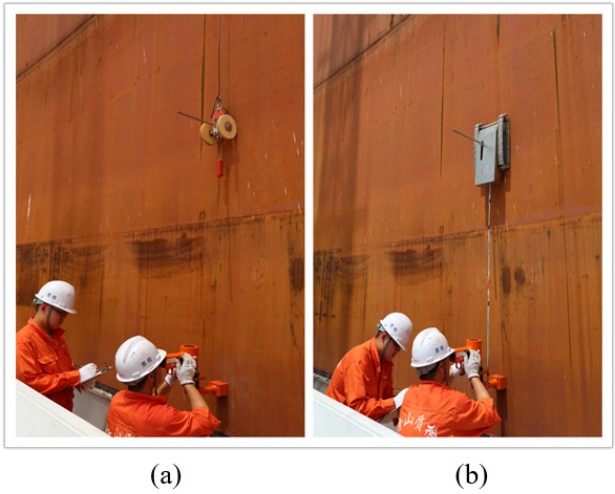

| (a) | (b) |

**Figure 14.** Internal measurement progress with (**a**) trolley and (**b**) TWCR.

**Table 5.** Radial deviation with trolley and tracked wall-climbing robot (TWCR) (internally).

| Ring Number | Radial Deviation with Trolley (mm) | Radial Deviation with TWCR (mm) | Difference (mm) |
|:---:|:---:|:---:|:---:|
| 1 | 5.4 | 5.4 | 0.0 |
| 2 | 1.3 | 1.3 | 0.0 |
| 3 | 0.0 | 0.0 | 0.0 |
| 4 | −2.2 | −1.1 | −1.1 |
| 5 | −1.1 | −0.2 | −0.9 |
| 6 | −1.5 | 0.3 | −1.8 |
| 7 | 0.0 | 1.1 | −1.1 |
| 8 | 0.2 | 1.6 | −1.4 |
| 9 | 1.0 | 2.7 | −1.7 |
| 10 | 2.0 | 3.8 | −1.8 |

It is noteworthy that the measurement performed by the manually-lifted trolley took 6 min per horizontal station, and the one performed by TWCR took 4 min per horizontal station. However, it was found to be problematical setting up power supply lines inside tanks.

### 3.3. External Measurement Test

The working principle of the external ORLM was introduced in Section 1. As is shown in Figure 15, an external floating roof tank with a nominal volume of 50,000 m$^3$, located in Zhoushan City, China,

was selected for the external experiment. On-site environmental conditions were wind speed 18km/h, temperature 30 °C, and relative humidity 65%.

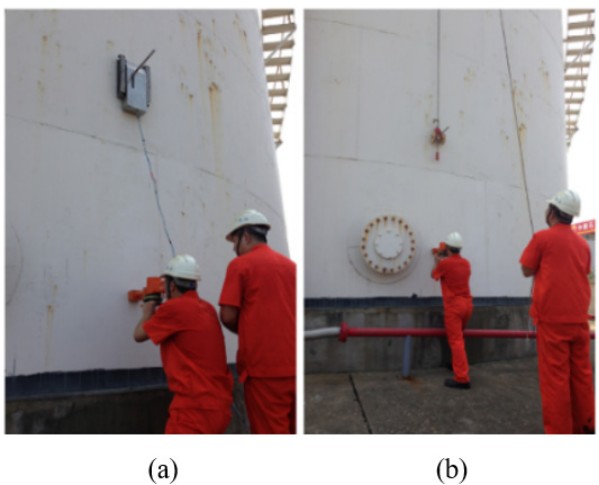

(a) (b)

**Figure 15.** External Measurement progress with (**a**) a TWCR and (**b**) trolley.

The reference circumference was 188,762 mm, and there are 62 horizontal stations and 18 vertical stations. The offsets measured by the two methods are presented in Table 6. The radial deviation in the table is the average of the deviations of the 62 horizontal stations in the same ring (each ring has two vertical stations).

**Table 6.** Radial deviation with trolley and WCR (externally).

| Ring Number | Radial Deviation with Trolley (mm) | Radial Deviation with TWCR (mm) | Difference (mm) |
|:---:|:---:|:---:|:---:|
| 1 | 0.0 | 0.0 | 0 |
| 2 | −0.7 | −0.2 | −0.5 |
| 3 | −1.8 | −1.2 | −0.6 |
| 4 | −4.3 | −3.8 | −0.5 |
| 5 | −10.1 | −9.2 | −0.9 |
| 6 | −7.5 | −6.3 | −1.2 |
| 7 | −2.7 | −2.0 | −0.7 |
| 8 | 1.8 | 0.9 | 0.9 |
| 9 | 15.5 | 14.2 | 1.3 |

It is noteworthy that the measurement performed by the manually-lifted trolley took 4 min per horizontal station, and the one performed by TWCR took 3 min per horizontal station.

*3.4. Results*

According to the JJG168-2005 (Verification Regulation of Vertical Metal Tank Capacity Sector/Industry), the maximum measurement tolerance of the radial deviation or radius of each ring is ±2.0 mm. It can be seen in Tables 5 and 6 that the deviation of the radial deviation from the actual standard value (radial deviation obtained by tradition) is less than 2.0 mm using the TWCR, both in the internal and external tests.

The raw field test data of the external measurement test were further processed to develop the volume table. A comparison of the volume data (1 m between the measuring points) acquired by traditional methods and the TWCR is presented in Table 7. It can be seen in Table 7 that,

$$\max\left(\left|\frac{y_1 - y_2}{(y_1 + y_2)/2}\right|\right) = 0.0042\%. \tag{14}$$

**Table 7.** Volume data comparison.

| Number | Height (m) | Volume (m$^3$) | | $\left\| \dfrac{y_1 - y_2}{(y_1 + y_2)/2} \right\|$ |
| | | Data Acquired by Trolley $y_1$ | Data Acquired by TWCR $y_2$ | |
|---|---|---|---|---|
| **1** | **2** | 5339.643 | 5339.643 | 0 |
| 2 | 3 | 8169.737 | 8169.663 | $0.009 \times 10^{-3}$ |
| 3 | 4 | 11,000.019 | 10,999.851 | $0.015 \times 10^{-3}$ |
| 4 | 5 | 13,830.800 | 13,830.526 | $0.020 \times 10^{-3}$ |
| 5 | 6 | 16,662.043 | 16,661.656 | $0.023 \times 10^{-3}$ |
| 6 | 7 | 19,493.592 | 19,493.098 | $0.025 \times 10^{-3}$ |
| 7 | 8 | 22,326.061 | 22,325.473 | $0.026 \times 10^{-3}$ |
| 8 | 9 | 25,158.531 | 25,157.848 | $0.027 \times 10^{-3}$ |
| 9 | 10 | 27,992.441 | 27,991.590 | $0.030 \times 10^{-3}$ |
| 10 | 11 | 30,826.383 | 30,825.361 | $0.033 \times 10^{-3}$ |
| 11 | 12 | 33,660.512 | 33,659.281 | $0.037 \times 10^{-3}$ |
| 12 | 13 | 36,494.719 | 36,493.258 | $0.040 \times 10^{-3}$ |
| 13 | 14 | 39,328.691 | 39,327.051 | $0.042 \times 10^{-3}$ |
| 14 | 15 | 42,162.371 | 42,160.594 | $0.042 \times 10^{-3}$ |
| 15 | 16 | 44,995.902 | 44,994.047 | $0.041 \times 10^{-3}$ |
| 16 | 17 | 47,828.730 | 47,827.043 | $0.035 \times 10^{-3}$ |
| 17 | 18 | 50,661.559 | 50,660.043 | $0.030 \times 10^{-3}$ |
| 18 | 19 | **53,492.402** | **53,491.121** | $0.024 \times 10^{-3}$ |

Note: As the lowest measuring point is in 2m, the 1m location is not used as a comparison point.

According to the JJG168-2005 (Verification Regulation of Vertical Metal Tank Capacity Sector/Industry), the expanded uncertainty in the capacity calibration of the tank with volumes greater than 700 m$^3$ should be less than 0.1%, and the compound uncertainty of two gauges can be expressed as follows:

$$\sqrt{U_1^2 + U_2^2} = \sqrt{0.1\%^2 + 0.1\%^2} = 0.14\%, \tag{15}$$

and

$$\max\left( \left\| \frac{y_1 - y_2}{(y_1 + y_2)/2} \right\| \right) < \sqrt{U_1^2 + U_2^2}. \tag{16}$$

Therefore, the results satisfy the requirements of the JJF1033-2016 (Rule for the examination of measurement standards). This tank was also measured by the liquid fill method (a more accurate method), and the measured volume at 19 m was 53,491.026 m$^3$, which is closer to the result of the TWCR (53,491.121 m$^3$) than the result of the trolley (53,492.402 m$^3$).

As it is not required for the operator to climb to the top of the tank and install the pully, the TWCR can reduce the time of the operation by 30%, as recorded, it requires only one operator to perform the capacity calibration, and the risk of operators falling while installing the pulley is removed.

## 4. Conclusions

In this study, a robot with climbing ability was proposed to replace the manually-lifted trolley, used in the traditional optical vertical line method. The permanent magnetic adhesion tracks enabled the TWCR to move steadily on the metal surface of the tanks without excessive energy costs. The scale used in the ORLM was designed to be collapsible to avoid collisions with the fire pipelines. A shovel-like rust remover could be added to remove the rust in front of the TWCR that could decrease the adhesion force.

Two sets of experiments (internal ORLM and external ORLM) were conducted, and the TWCR exhibited significant adaptability to the complex conditions of the tank surfaces. The experiments also indicated that the capacity calibration method using the TWCR is highly accurate, reduces the operating time, and operating hazards compared to the traditional ORLM.

However, it was found to be problematical setting up power supply lines inside tanks. We expect that, by changing the power supply to a battery, the applicable working conditions of the TWCR could be expanded.

**Author Contributions:** The work described in this article is the collaborative development of all authors. Conceptualization, X.C. and H.H. (Haocai Huang); methodology, Y.W.; investigation, H.H. (Huadong Hao); data curation, H.S.; supervision, H.H. (Haocai Huang); writing—original draft preparation, Y.W.; writing—review and editing, X.C. and H.H. (Haocai Huang).

**Funding:** The study was supported financially by the Science and Technology Project of AQSIQ (Grant No. 2017QK075) and the Quality Technology Infrastructure Project of Zhejiang Quality Supervision System (Grant No. 20180130).

**Acknowledgments:** The authors would like to thank Zenan Wu, Zhenqian Shen, and other staff of the Zhoushan Institute of Calibration and Testing for Quality and Technology Supervision for performing the experiments.

**Conflicts of Interest:** The authors declare no conflict of interest.

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
