# Peer review of "Tracked Wall-Climbing Robot for Calibration of Large Vertical Metal Tanks"

_applsci, doi:10.3390/app9132671_

Round 1
Reviewer 1 Report
The paper presents an interesting wall-climbing robot with permanent magnetic adhesion tracks. The quality of the paper is sufficient. Following is a list of comments to help the authors to improve the paper quality.
1) If it is possible provide more details of the control algorithm/system.
2) Provide results comparisons with similar systems (if any) (noncompulsory)
3) A 30W AC gear motor with a reduction ratio of 1:36 installed on the baseboard delivers power to the gear train comprising 2 electromagnetic clutches. It may be useful a simple formula to be provided of the power concluded that is required to overcome the adhesive power in order for the robot to move forward/backward.
Author Response
Dear Reviewer,
Thanks very much for your constructive and inspiring comments. They are invaluable in assisting our paper writing and in guiding our further academic research and are highly appreciated. We have further revised the paper according to your suggestions. Please allow us to explain the revisions to you one by one as follows:
1) If it is possible provide more details of the control algorithm/system.
Answer: Thanks very much for your inspiring suggestion. In the revised version, a diagram of the PID controller has been added as well as the input/output of it.
2) Provide results comparisons with similar systems (if any) (noncompulsory)
Answer: There are some similar systems but we could not find their field tests and the tests’ data, that’s why we didn’t present result comparisons.
3) A 30W AC gear motor with a reduction ratio of 1:36 installed on the baseboard delivers power to the gear train comprising 2 electromagnetic clutches. It may be useful a simple formula to be provided of the power concluded that is required to overcome the adhesive power in order for the robot to move forward/backward.
Answer: Thanks very much for your constructive suggestion. In the revised version, the necessary torque of the motor has been estimated and we have taken the torque of the gravity and the torque of the lowest magnetic unit which is disengaging with the surface into consideration.
Reviewer 2 Report
In this paper a wall-climbing robot, equipped with magnetic tracks, for the volume calibration of large vertical metal tanks is presented. In particular, the robot has to climb the tanks walls and moves at different and certain heights from ground; thanks to a target scale mounted on it, the radius deviation is measured by an optical device sited at ground.
Compared to conventional system (i.e. a trolley driven by a pulley), this method lead to a significant time saving.
Authors describe the design of the robot structure, magnetic tracks and transmission, as well as of the mechanism of the foldable and adjustable target scale. Moreover, the prototype is shown and the results of some measuring tests on real tanks are reported and compared with conventional methods.
The paper is well-written and well-match with the journal topic. In my opinion it can be accepted with minor revisions.
Remarks and suggestions:
- rows 270, 280 and 286 and table 7: use consistent number representation (same thousand separator).
- From the text is not clear how the initial alignment of the robot with the vertical axis is executed. Do you aligned the robot manually, or you exploit the robot steer? And in this second case, how much precise are the two clutches in performing this kind of motion? You should explain better in the text.
- In table 5 and 6 you present the comparison between measures done with your tracked system and conventional system (i.e. trolley). You should consider a mean of different measures in order to reduce the human error, at least for the hand-made ones. In fact the differences you reported on the fourth column can be due also to an error in the conventional measure. The mean reduces this problem.
- Section "Control system" lacks of information. For example is not clear if the control system adjust by itself the robot height from ground by reading the position sensor, or if the operator have to do this manually.
- Tracked vehicle generally use two different motors for the traction. You could better explain why you move on this kind of solution (i.e. single motor+clutches) and which are the pros and cons.
- Why you decided to operate with a 220V main supply in a risky working environment
Author Response
Dear Reviewer,
Thanks very much for your constructive and inspiring comments. They are invaluable in assisting our paper writing and in guiding our further academic research and are highly appreciated. We have further revised the paper according to your suggestions. Please allow us to explain the revisions to you one by one as follows:
1) rows 270, 280 and 286 and table 7: use consistent number representation (same thousand separator).
Answer: Thanks very much for your constructive suggestion. In the revised version, the number representation is consistent.
2) From the text is not clear how the initial alignment of the robot with the vertical axis is executed. Do you aligned the robot manually, or you exploit the robot steer? And in this second case, how much precise are the two clutches in performing this kind of motion? You should explain better in the text.
Answer: Thanks very much for your inspiring suggestion. In the revised version, we added the details of the beginning of the experiments, which is:
The robots were carefully placed on the metal surface and aligned vertically, after which the TWCR itself further adjusted the attitude bias (if any).
3)In table 5 and 6 you present the comparison between measures done with your tracked system and conventional system (i.e. trolley). You should consider a mean of different measures in order to reduce the human error, at least for the hand-made ones. In fact the differences you reported on the fourth column can be due also to an error in the conventional measure. The mean reduces this problem.
Answer: Thanks very much for your inspiring suggestion. The description of the data in the table 5 and 6 was not clear and we have added the sources of the data in the revised version. Actually, the radial deviation in the table is the average of the deviations of the 60+ horizontal stations in the same ring (each ring has 2 vertical stations). It may reduce the human error.
However, because the experiment takes a long time, we did have only one reading at a single measuring point, which may lead to the human error. We will take two measurements at a single measuring point in the further research.
4)Section "Control system" lacks of information. For example is not clear if the control system adjust by itself the robot height from ground by reading the position sensor, or if the operator have to do this manually.
Answer: Thanks very much for your constructive suggestion. In the revised version, a diagram of the PID controller has been added as well as the input/output of it. We have made it clear that the height of the TWCR is visually controlled by the operator according to the distance from the adjacent horizontal weld seams, and the precise position of the scale is measured by the optical device and manually adjusted by the operator through the scale-moving mechanism. And we mistakenly wrote the attitude sensor as a position sensor, it has been corrected now.
5)Tracked vehicle generally use two different motors for the traction. You could better explain why you move on this kind of solution (i.e. single motor+clutches) and which are the pros and cons.
Answer: Thanks very much for your inspiring suggestion. In the revised version, we have added the explanation of the solution:
Compared to the use of two motors for traction, a single motor with two clutches, though the presence of clutches introduces extra power consumption and transmission loss, make sufficient room for the calibration system on the compact body. The torque transmitted by the electromagnetic particle clutch is linear with the control current, which means the torque can be easily controlled.
6)Why you decided to operate with a 220V main supply in a risky working environment
Answer: In the revised version, we have made it clear that capacity calibrations are conducted before the first use or after maintenance, when the tanks are empty and free from the risk of explosion, thus the robot is powered by a wire delivering 220 V AC, which is readily available close to the tanks.
Reviewer 3 Report
This paper presents a tracked wall-climbing robot (TWCR) for the calibration of large vertical metal tanks. The TWCR allows to perform more accurate, faster and more safe calibrations operations with respect to the traditional method usually adopted.
The paper is well written, easy to read and the English is good. The methods and the prototype are well described, the experimental tests are properly commented and analyzed.
I have some minor comments that the authors should address to improve the quality of the manuscript.
1. I would suggest the authors to add one column in Table 1 to include some examples of references in which each of the four methods for capacity calibration are deeply explained.
2. Lines 67-73. I suggest the authors to spend few more lines to analyze and explain the different adhesion and locomotion systems adopted for climbing robots.
3. Tables 2 and 3. As for Table 1, I would suggest the authors to add one column with some examples of references for each type of adhesion and locomotion mechanism.
4. The classification of climbing robots can be improved by adding some more examples of wall climbing robots (recently published), such as:
“Upside-Down Robots: Modeling and Experimental Validation of Magnetic-Adhesion Mobile Systems”, Robotics 8 (2), 41; https://doi.org/10.3390/robotics8020041,
in which a novel class of climbing robots based on a sandwich configuration and capable of climbing non-ferromagnetic surfaces is presented.
5. Section 2.5. Please spend few more lines to describe the control algorithm. What are the input and output signals of the PID controller? How was the PID controller tuned?
Some minor remarks:
6. Figure 2. Please do not split the figure between two pages. (If not possible, please add a separate label to the first image of figure 2).
7. Line 112. “It shall have five basic functions”. Actually, the bullet points are 6. Please correct.
8. Line 117. “The linear motion deviation shall be less than 1°50′ from the perpendicular “. How did you choose this value? Please briefly explain.
I suggest the paper to be accepted after minor revision.
Author Response
Dear Reviewer,
Thanks very much for your constructive and inspiring comments. They are invaluable in assisting our paper writing and in guiding our further academic research and are highly appreciated. We have further revised the paper according to your suggestions. Please allow us to explain the revisions to you one by one as follows:
1. I would suggest the authors to add one column in Table 1 to include some examples of references in which each of the four methods for capacity calibration are deeply explained.
Answer: Thanks very much for your constructive suggestion. In the revised version, we have added the references of the four methods.
2. Lines 67-73. I suggest the authors to spend few more lines to analyze and explain the different adhesion and locomotion systems adopted for climbing robots.
Answer: Thanks very much for your constructive suggestion. In the revised version, we have added some description of the principle of the adhesion systems as well the pros and cons.
3.Tables 2 and 3. As for Table 1, I would suggest the authors to add one column with some examples of references for each type of adhesion and locomotion mechanism.
Answer: Thanks very much for your constructive suggestion. In the revised version, we have added the references in the tables 2 and 3.
4. The classification of climbing robots can be improved by adding some more examples of wall climbing robots (recently published), such as:
“Upside-Down Robots: Modeling and Experimental Validation of Magnetic-Adhesion Mobile Systems”, Robotics 8 (2), 41; https://doi.org/10.3390/robotics8020041,
in which a novel class of climbing robots based on a sandwich configuration and capable of climbing non-ferromagnetic surfaces is presented.
Answer: Thanks very much for your inspiring suggestion. The article you mentioned is very innovative and we have added it as a new kind of wall climbing robot.
5. Section 2.5. Please spend few more lines to describe the control algorithm. What are the input and output signals of the PID controller? How was the PID controller tuned?
Answer: Thanks very much for your suggestion. In the revised version, a diagram of the PID controller has been added as well as the input/output of it. We also added the description of the tuning of the PID controller.
6. Figure 2. Please do not split the figure between two pages. (If not possible, please add a separate label to the first image of figure 2).
Answer: Thanks very much for pointing out this. We have made the figures in a single page in the revised version.
7. Line 112. “It shall have five basic functions”. Actually, the bullet points are 6. Please correct.
Answer: Thanks very much for pointing out this. We have corrected it in the revised version.
8. Line 117. “The linear motion deviation shall be less than 1°50′ from the perpendicular “. How did you choose this value? Please briefly explain.
Answer: In the revised version, we have made it clear that:
The linear motion deviation shall be less than 1°50′ from the perpendicular to guarantee that the scale is visible by the optical device during the calibration of a single horizontal station.